# Graph Contrastive Learning via Interventional View Generation

## ABSTRACT

Graph contrastive learning (GCL), as a popular self-supervised learning technique, has demonstrated promising capability in learning discriminative representations for diverse downstream tasks. A large body of GCL frameworks mainly work on graphs formed under *homophily effect*, i.e., similar nodes tend to connect with each other. In their design, the augmentation and aggregation are usually conducted indiscriminately on edges, *ignoring the existence of heterophilic edges that connect dissimilar nodes*. Therefore, the efficacy of GCL could greatly deteriorate on heterophilic graphs, verified by our analysis: GCL on a mixture of homophilic and heterophilic edges will generate representations that are indistinguishable across different classes in the embedding space. To address this challenge, we propose a novel GCL framework via *interventional view generation*. Specifically, we generate homophilic and heterophilic views through counterfactual intervention, which targets on disentangling homophilic and heterophilic structure from the original graph, such that we can capture their corresponding information using separate filters in the contrastive learning process. Since the homophilic view and the heterophilic view present different frequency signals, they are further encoded via a low-pass and a high-pass filter respectively. Extensive experiments on multiple benchmark datasets demonstrate the effectiveness of our design. Our proposed framework achieves a remarkably improved downstream performance on graphs with high heterophily while maintaining a comparable ability in learning homophilic graphs. A comprehensive study also verifies the necessity of individual designs in our framework.

## KEYWORDS

Graph Contrastive Learning, Heterophilic Graph Learning, Counterfactual Analysis

**ACM Reference Format:**

Anonymous Author(s). 2018. Graph Contrastive Learning via Interventional View Generation. In *Proceedings of Make sure to enter the correct conference title from your rights confirmation emai (Conference acronym 'XX)*. ACM, New York, NY, USA, 10 pages. https://doi.org/XXXXXXX.XXXXXXX

## 1 INTRODUCTION

Graphs neural networks (GNNs) have attracted tremendous attention due to their superior capability in modeling graph-structured data, such as social networks [21, 25], traffic networks [33], and recommendation systems [17, 37]. GNNs commonly adopts the

message passing mechanism to iteratively update the node representations via neighbor aggregation, thus can effectively capture both node feature and graph structure. While the message passing mechanism introduces neighboring context to help node representation learning, its performance is highly related to the *homophily* assumption, which states that connected nodes should be similar.

However in practice, such assumption does not always hold. In fact, many real-world web-based graphs are highly *heterophilic*, and in such graphs, connected nodes are highly likely from different classes or have different features [19, 42]. For instance, in online dating networks, users are more likely to connect with people of opposite gender; in online transaction graphs, fraudsters tend to connect with normal customers instead of other fraudsters [19]. Due to the ubiquity of graph heterophily in modern web applications and systems, models that excel under homophily assumption should adapt to the presence of heterophily property. However, directly applying the message passing mechanism upon such heterophilic graphs could be problematic: the aggregated information of dissimilar local neighbors could greatly deteriorate the node representation quality and further influence the downstream performance.

To tackle the issue introduced by heterophily, enormous solutions have been proposed. One line of works focuses on *GNN architecture* tailored for heterophilic graphs [10, 16, 20, 24, 31, 43], including feature and structure separation[10], ego and neighbor separation[10, 31], high-order neighbor mixup[16, 43],message aggregation adjustment[16] and neighbor reweighting [16, 20]. Nevertheless, these solutions typically rely on expert knowledge to manually adjust the model design based on the degree of heterophily. Considering the complex and diverse patterns exist in different graphs, such heuristic designs may not generalize across graphs and could require heavy human efforts for model tuning. Another line of research aims to save human efforts via *automated GNN learning* [40] based on neural architecture search (NAS). Through well-designed search space and efficient search strategies, these methods can automatically find the best model architecture on a given dataset. However, such solution require classification performance as reward, thus can only serve for the supervised learning setting.

Until very recently, a few works start to investigate heterophily in unsupervised setting [3, 15, 26, 34], without accessing any downstream label information. In the graph *auto-encoder learning* framework, heterophiliy is handled through either neighbor reconstruction [26] or decoupled graph generative assumption [34]. In the graph *contrastive learning* framework, existing solutions estimate heterophilic edges via node similarity [3] or edge similarity [15], which could lead to suboptimal performance without principled guidance.

In this paper, we focus on designing a graph *contrastive learning* framework that can adapt to different degree of heterophily. Our method is motivated by a preliminary analysis, from which we find two essential designs for successful node representation

learning on heterophilic graphs: seperation between heterophilic and homophilic edges, and a specialized graph filter for each type of edges. We further integrate them into the graph contrastive learning framework. Specifically, we propose a counterfactual solution to disentangle heterophilic patterns from homophilic ones, such that by controlling the treatment factor, we can generate *interventional homophilic and heterophilic views* from the original graph. These two views will be fed into different graph filters: a *low-pass filter* is used to encode the homophilic view, while a *high-pass filter* is for the heterophilic view. This design is inspired by recent findings [2, 14, 16] that low-pass filters focus on smoothly varying signals on similar neighbors (in the homophilic view), and high-pass filters capture sharply changing signals on dissimilar neighbors (in the heterophilic view). Finally, we contrast the embedding from both views following a common contrastive learning practice.

Compared with developing GNN architecture tailored for heterophilic graphs [10, 16, 20, 24, 31, 43], our framework does not require any complicated architecture designs; and instead of discriminating heterophilic edges curtly based on node similarity or edge similarity [3, 15], our counterfactual design enjoys a principled guidance from the treatment factor. Extensive experiments on both homophilic and heterophilic graphs verify the effectiveness of our proposed framework. In summary, our contribution can be highlighted as follows:

- We propose a principled solution to disentangle homophily and heterophily from the counterfactual perspective, which can guide homophilic and heterophilic vew generation with intervention;
- We integrate interventional view generation and distinct view filters in a normal contrastive learning pipeline, without complicated model architecture or learning framework designs;
- We demonstrate the improvement by the proposed method and the effectiveness of each design on multiple homophilic and heterophilic benchmarks.

## 2 RELATED WORK

### 2.1 GNNs with Heterophily

To better adapt to graph heterophily and mitigate over-smoothing issues, recent approaches have proposed a set of alternative aggregation mechanisms, focusing on tailoring GNN architectures [10, 16, 20, 24, 31, 43], encompassing structure separation [10], ego and neighbor separation [10, 31], high-order neighbor mixup [16, 43], specialized message passing [16] and neighbor reweighting [16, 20]. For instance, UGCN [10] constructs neighbor sets by selecting nodes based on their feature similarity. GGNN [31] maps ego and neighbor feature into separate representations. H2GCN [43] employs specialized GNN designs including ego/neighbor separation and high-order neighbor mixup. ACM [16] modifies message passing via high- and low-pass filter, and identity channels to diversify local information. Another line of research designs automated GNN learning paradigm [40] to avoid extensive manual model tuning. These methods can automatically discover the optimal model architecture for a given dataset through a well-designed search space and effective search strategies. However, such a solution requires classification performance as the reward, which is typically applicable only in a supervised learning setting.

## 2.2 Self-Supervised Learning with Heterophily

Graph self-supervised learning aims to pretrain a feature extractor on unlabeled dataset via graph reconstruction (as in auto-encoder framework [7, 12, 13, 26]) or instance discrimination (as in contrastive learning [8, 23, 28, 30, 38, 44, 45]). *Graph autoencoder* framework learns informative representations by recovering the feature and structure pattern in the original graph. *Graph contrastive learning* (GCL) learns discriminative representations by maximizing mutual information between positive views contrast to negative views. For instance, GCA [45], GRACE [44], and GraphCL [38] extend SimCLR [4] to learn node or graph representations by contrasting either in node level or graph level. DGI [30], InfoGraph [23], and MVGRL [8] adopt cross-level contrast. BGRL [28] performs view-level representation prediction without negative samples.

These vanilla self-supervised methods usually use low-pass GNN backbones encouraging smoothness between neighbors, thus could lead to suboptimal performance on heterophilic graphs. Until recently, people start to investigate the heterophily issue in the *graph autoencoder* framework [26, 34, 41] and the *graph contrastive learning* framework [3, 15, 31, 35]. Our focus in this paper is graph contrastive learning with heterophily. For sample, HGRL [3] drops low-similarity pairs and SP-GCL [31] treats them as negative pairs, which however could also lose useful information of these heterophilic edges. GREET [15] directly learns non-homophilic edges based on node features. These heuristics estimates heterophilic edges without proper guidance, which could be inaccurate and mislead the model. HLCL [35] directly uses high-and low-pass filters on views of the original graph, and could be ineffective if the views have mixed edges. In this work, we aim to design a principled homophilic/heterophilic view generation method guided by counterfactual treatment, such that specialized filters can trained to capture distinct patterns in each view.

## 3 PRELIMINARY

We first briefly introduce the background of graph neural networks (GNNs), as well as graph contrastive learning (GCL). And then we provide a preliminary analysis to demonstrate the necessity of separating heterophilic and homophilic structure in GCL for achieving high-quality node representations.

### 3.1 Background

*3.1.1 Graph.* Define a graph as $\mathcal{G} = (\mathcal{V}, \mathcal{E})$, where $\mathcal{V} = \{v_1, \ldots, v_n\}$ represents the node set and $\mathcal{E} \subseteq \mathcal{V} \times \mathcal{V}$ represents the edge set. The edge connecting node $v_i$ and $v_j$ is denoted as $e_{i,j}$. The numbers of nodes and edges are denoted as $|\mathcal{V}| = n$ and $|\mathcal{E}| = m$, respectively. Let $X \in \mathbb{R}^{n \times d_f}$ denote the feature matrix, where the $i$-th row $\mathbf{x}_i$ represents the $d_f$-dimensional feature vector of node $v_i$. We represent $\mathcal{G}$'s adjacency matrix as $A \in \mathbb{R}^{n \times n}$, where $A_{ij} = 1$ if $e_{i,j} \in E$ and $A_{ij} = 0$ otherwise. Using the feature matrix and adjacency matrix, the graph can also be written as $\mathcal{G} = (A, X)$. The symmetric normalized adjacency matrix is denoted as $\tilde{A} = D^{-1/2} A D^{-1/2}$, where $D$ is the diagonal degree matrix such that $D_{ii} = \sum_j A_{ij}$. The Laplacian matrix of the graph is defined as $L = D - A$, and the symmetric normalized Laplacian matrix is $\tilde{L} = I - \tilde{A}$.

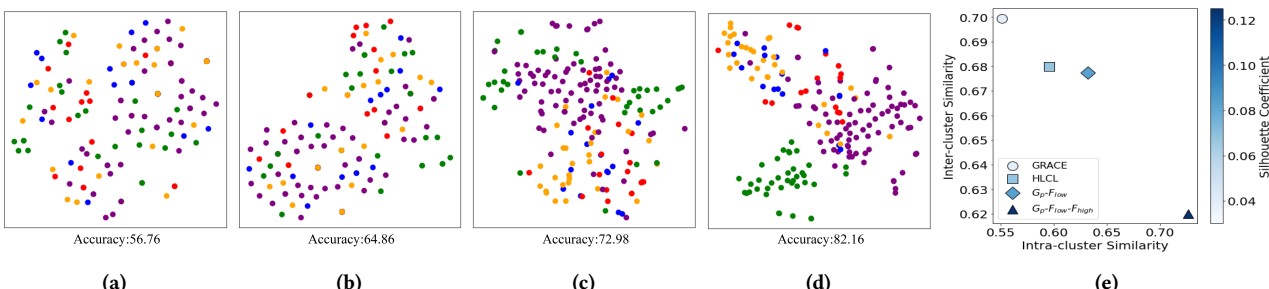

**Figure 1: Visualization of node representations obtained by GCL variants on Cornell: (a). GRACE; (b). HLCL; (c). $f_{low}(\mathcal{G}^{homo^*}) \oplus f_{low}(\mathcal{G}^{heter^*})$; (d). $f_{low}(\mathcal{G}^{homo^*}) \oplus f_{high}(\mathcal{G}^{heter^*})$. Their clustering performance is quantitatively evaluated in (e).**

### 3.1.2 Graph Neural Network (GNN).

Denote the neighbor set of node $v_i$ as $N(i) = \{v_j \mid (v_i, v_j) \in \mathcal{E}\}$. GNNs are designed to generate representations of each node by passing and aggregating its neighboring nodes. Formally, for node $v_i$, a general message-passing scheme updates its representation as follows:

$$\mathbf{z}_i^{(k)} = \text{UPDATE}(\mathbf{z}_i^{(k-1)}, \text{AGGREGATE}(\mathbf{z}_j^{(k-1)} \mid j \in N(i))), \quad (1)$$

where $\text{AGGREGATE}(\cdot)$ and $\text{UPDATE}(\cdot)$ are trainable functions used for neighbor aggregation and representation update. For instance, **SGC** [32] designs a *low-pass filter*, which updates the representations for all nodes by $Z^{(k)} = \tilde{A} Z^{(k-1)} W^{(k-1)}$ to encourage neighbor smoothness. Correspondingly, the *high-pass filter* can be formulated as $Z^{(k)} = \tilde{L} Z^{(k-1)} W^{(k-1)}$ to capture neighbor dissimilarity. And $W^{(k-1)}$ is the model parameter for layer $k-1$.

### 3.1.3 Graph Contrastive Learning (GCL).

As a popular self-supervised learning paradigm, GCL aims to pretrain a GNN encoders in absence of supervision. It realizes instance discrimination via the InfoMax principle [9], which maximizes the mutual information between positive samples. Positive samples are typically generated by graph augmentation. Based on the assumption that augmentation does not ruin data semantics, augmented views from the same graph are considered as positive samples, while views from different graphs are negative samples. A GNN model then encodes views and is pretrained to minimize the commonly used InfoNCE loss [18], defined on each node as follows:

$$\mathcal{L}_{\text{GCL}}(\mathbf{z}_i^{\mathcal{G}_1}, \mathbf{z}_i^{\mathcal{G}_2}) = -\log \frac{\exp(\text{sim}(\mathbf{z}_i^{\mathcal{G}_1}, \mathbf{z}_i^{\mathcal{G}_2})/\tau)}{\sum_{j \neq i} \exp(\text{sim}(\mathbf{z}_i^{\mathcal{G}_1}, \mathbf{z}_j^{\mathcal{G}_2})/\tau)}, \quad (2)$$

where $\mathbf{z}_i^{\mathcal{G}_1}$ and $\mathbf{z}_i^{\mathcal{G}_2}$ are embeddings of node $v_i$ in two augmented views $\mathcal{G}_1$ and $\mathcal{G}_2$ respectively, $\text{sim}(\cdot, \cdot)$ is a similarity metric (i.e., cosine similarity), and $\tau$ denotes the temperature parameter. By minimizing the contrastive loss, the embeddings of the same node in different views are pushed close, while the embeddings of different nodes will dispart, such that node discrimination is achieved.

### 3.1.4 Graph Heterophily.

The degree of homophily in a graph can be quantified by two metrics: node homophily [20] and edge homophily [43]. Specifically, node homophily is defined as the average proportion of neighbors belonging to the center node's class:

$$H_{\text{node}} = \frac{1}{|\mathcal{V}|} \sum_{v_i \in \mathcal{V}} \frac{|\{v_j \in N(i) \mid y_i = y_j\}|}{|N(i)|}, \quad (3)$$

where $y_i$ denotes the ground truth label of node $v_i$. The other measure, edge homophily, calculates the proportion of homophilic edges (i.e., connecting nodes of the same class) in the whole graph:

$$H_{\text{edge}} = \frac{|\{(v_i, v_j) \in \mathcal{E} \mid y_i = y_j\}|}{|\mathcal{E}|}. \quad (4)$$

The values of both $H_{\text{node}}$ and $H_{\text{edge}}$ are within the range of $[0, 1]$. Graphs exhibiting strong homophily tend to have higher values of $H_{\text{node}}$ and $H_{\text{edge}}$ (i.e., close to 1), and graphs with strong heterophily will have lower $H_{\text{node}}$ and $H_{\text{edge}}$ (i.e., close to 0).

## 3.2 GCL with Heterophily

We conduct a preliminary analysis on a heterophilic graph to showcase how GCL can be tailored to adapt heterophily. In this analysis, we considered four GCL variants:

- *GRACE* [44], which is a vanilla GCL method that generates random augmented views and uses the same low-pass GCN encoder;
- *HLCL* [35], which generates random augmented views, but uses a low-pass and a high-pass filter to encode two views separately;
- $f_{low}(\mathcal{G}^{homo^*}) \oplus f_{low}(\mathcal{G}^{heter^*})$, which first separate homophilic view $\mathcal{G}^{homo^*} = \{(v_i, v_j) \in \mathcal{E} \mid y_i = y_j\}$ and heterphilic view $\mathcal{G}^{heter^*} = \{(v_i, v_j) \in \mathcal{E} \mid y_i \neq y_j\}$ based on node label supervision. Both views are encoded via a SGC [32] low-pass filter $f_{low}(\cdot)$;
- $f_{low}(\mathcal{G}^{homo^*}) \oplus f_{high}(\mathcal{G}^{heter^*})$, which separates views in a similar way, but adopts different filters: a low-pass encoder for the homophilic view and a high-pass one for the heterophilic view.

The rest GCL designs (e.g., contrastive loss) keep the same across the variants. We run these GCL variants on the Cornell dataset with strong heterophily, and analyze the resulting node representations.

**Results** Figure 1 measures the clustering quality of node representations obtained by these variants. Intuitively in the 2D visualization, $f_{low}(\mathcal{G}^{homo^*}) \oplus f_{low}(\mathcal{G}^{heter^*})$ allows nodes to cluster better than GRACE and HLCL, and $f_{low}(\mathcal{G}^{homo^*}) \oplus f_{high}(\mathcal{G}^{heter^*})$ achieves the best clustering effect. Meanwhile in Figure 1e, we quantitatively measure the clustering performance: a good clustering result should have higher *intra-cluster similarity* and *silhouette coefficient*, with lower *inter-cluster similarity*. Therefore, the best clustering should locate at the bottom right of Figure 1e with a darker

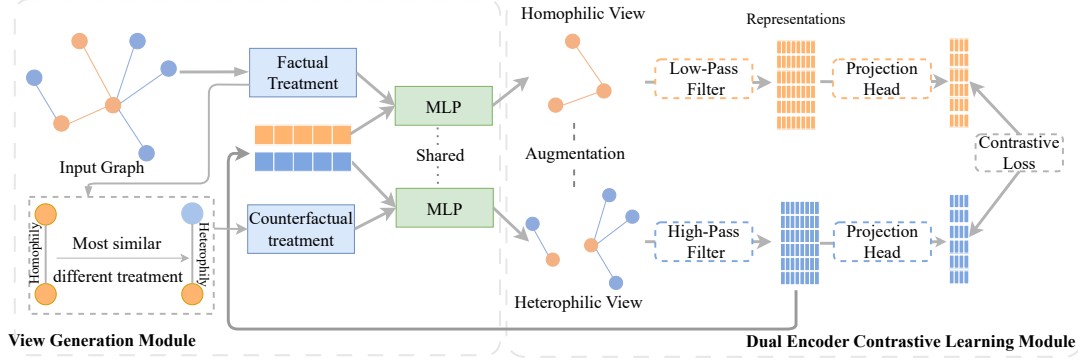

**Figure 2: Overview of GCL-IVG.** *View Generation Module* aims to generate homophilic and heterophilic views by counterfactual intervention. *Dual Encoder Contrastive Learning Module* applies low- and high-pass filters to encoder each view respectively.

color. We observe the ranking of clustering performance as follows:
$f_{low}(\mathcal{G}^{homo^*}) \oplus f_{high}(\mathcal{G}^{heter^*}) > f_{low}(\mathcal{G}^{homo^*}) \oplus f_{low}(\mathcal{G}^{heter^*}) >$ HLCL > GRACE.

**Remarks** Based on our observation in Figure 1, we identify two important designs in GCL that could lead to better node representation learning with heterophily:

- *Separation of homophilic and heterophilic edges*: the methods that separate these two types of edges achieve better clustering performance. This suggests that in the design of GCL, we should construct distinct views to separate homophilic and heterophilic structure from the original graph. Note that label information used here is just proof-of-concept. In practical GCL, this is a more challenging task with the absence of supervision.
- *Specialized low- and high-pass filter:* we and prior works [16, 43] confirm that low-pass filters encouraging similarity of neighbors are useful for homophilic views, while high-pass filters imparting neighbor distinctiveness better fit heterophilic networks. This suggests that in GCL, specialized filters should be adopted to encode homophilic/heterophilic views.

These two designs together play an integral role. Without view separation, only using different filters on randomly augmented views (i.e., HLCL) can not bring obvious benefit. And using the same low-pass filter on separated views (i.e., $f_{low}(\mathcal{G}^{homo^*}) \oplus f_{low}(\mathcal{G}^{heter^*})$) cannot achieve the same level of performance as using different filters. Motivated by this analysis, we propose to realize and integrate these two designs in GCL. Specifically, we target on designing a unsupervised view generator to separate homophilic and heterophilic structures, and then applying tailored filters for each view.

## 4 METHODOLOGY

**Overview** This section elaborates our proposed method, Graph Contrastive Learning via Interventional View Generation (GCL-IVG). Figure 2 illustrates the overview of GCL-IVG, which primarily comprises two modules: the View Generation Module and the Dual Encoding Contrastive Learning Module. In the View Generation Module, we decouple heterophilic and homophilic patterns through counterfactual methods. By controlling the treatment factor, interventional homophilic and heterophilic views can be generated from

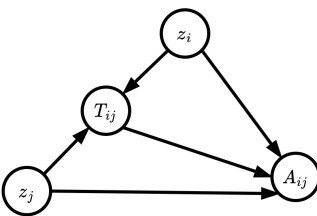

**Figure 3: In our causal model, given *node context* $(z_i, z_j)$ and *treatment* $T_{ij}$, we observe *factual adjacency outcome* $A_{ij}$.**

the original graph. In the Dual Encoding Contrastive Learning Module, a low-pass and a high-pass filter are employed to encode the homophilic and heterophilic view respectively. These two modules are naturally integrated in GCL framework, and can be optimized in an alternating manner. We now introduce each design in details.

### 4.1 Interventional View Generation

To successfully disentangle homophilic and heterophilic patterns from the original graph, we propose an *counterfactual intervention* method. By investigating the graph generative process, we aims to estimate the effect of heterophily treatment on the edge probability between two nodes, such that we can intervene the view generation by manipulating the heterophily treatment.

*4.1.1 Causal Model.* We leverage counterfactual causal inference to find out the causal relationship between treatment and outcome by asking a counterfactual question: *would the graph structure change if the treatment was different?* We assume the observed graph structure is generated following the causal relationship [39], which is depicted in Figure 3. Given the context, treatments and corresponding outcomes, counterfactual inference methods target on finding the effect of treatment on the outcome. Here, the information on node $v_i$ and $v_j$ is the *context*, denoted as $(z_i, z_j)$. We denote by $A$ the observed adjacency matrix as the *factual outcomes*, and denote by $A^{CF}$ the unobserved matrix if the treatment is different as the *counterfactual outcomes*. We represent the binary *treatment* matrix as $T \in \{0, 1\}^{n \times n}$, where $T_{ij}$ implies the treatment to the

node pair $(v_i, v_j)$. Correspondingly, the *counterfactual treatment* matrix can be denoted as $T^{CF}$, where $T_{ij}^{CF} = 1 - T_{ij}$.

*4.1.2 Treatment Variable.* We define the treatment $T_{ij}$ as the homophily property of node pair $(v_i, v_j)$, i.e., whether these two nodes should present homophily. For $T_{ij} = 1$, it suggests edge existence between similar node pair $(v_i, v_j)$ and inexistence between dissimilar $(v_i, v_j)$. Let $c : \mathcal{V} \to \mathbb{N}$ denote a partition strategy based on node context information. We understand homophily as two nodes belonging to the same partition, i.e., $T_{ij} = 1$ if $c(v_i) = c(v_j)$, and $T_{ij} = 0$ otherwise. Empirically, we realize this partition oracle as a condition on the node context, which judges whether the feature similarity between node pair $(v_i, v_j)$ exceeds the average similarity of all node pairs in the graph.

*4.1.3 Counterfactual Outcome.* We now introduce the generation of counterfactual outcome. Since for each node pair, we can only observe the factual treatment and corresponding outcomes, the relationship between node pairs with an opposite treatment is unknown. Therefore, we estimate the counterfactual outcome as the outcome by the closest observed context[1]. In other words, given a set of observed node pairs, we aim to find their nearest neighbors with the opposite treatment, and regard them as counterfactual node pairs. For each node pair $(v_i, v_j)$, we define its counterfactual relationship as:

$$(v_a, v_b) = \arg\max_{v_a, v_b \in \mathcal{V}} \left[ s((v_i, v_j), (v_a, v_b)) | T_{ab} = 1 - T_{ij} \right], \quad (5)$$

where $s(\cdot, \cdot)$ is a similarity measurement between a pair of contexts. Following [39], we conduct node-wise comparing to realize the context comparison. Specifically, we estimate the counterfactual outcome for for each node pair $(v_i, v_j)$ as:

$$(v_a, v_b) = \arg\max_{v_a, v_b \in \mathcal{V}} \left[ s(\tilde{\mathbf{x}}_i, \tilde{\mathbf{x}}_a) + s(\tilde{\mathbf{x}}_j, \tilde{\mathbf{x}}_b) \mid T_{ab} = 1 - T_{ij} \right], \quad (6)$$

where $s(\cdot, \cdot)$ is implemented as the Euclidean similarity in the embedding space of $\tilde{X}$. Empirically, the embeddings $\tilde{X}$ are obtained by concatenating nodes' original feature and their structural encoding (via Node2Vec [7]), which is less influenced by homophily. Enumerating over all $O(n^2)$ node pairs to solve this combinatorial optimization problem takes $O(n^4)$ comparisons, which is extremely infeasible in practice. Therefore, we apply a downsampling strategy to sample a set of unconnected node pairs and combine with all connected node pairs as the outcome set. Ultimately, we can obtain the counterfactual outcome $A^{CF}$ as follows:

$$A_{ij}^{CF} = \begin{cases} A_{ab} & \text{, if } \exists (v_a, v_b) \in \mathcal{V} \times \mathcal{V} \text{ satisfies Eq. (6)}; \\ A_{ij} & \text{, otherwise.} \end{cases} \quad (7)$$

Note that the counterfactual outcome $A^{CF}$ can be computed in the pre-processing step, and will be fixed in the later stage.

*4.1.4 Interventional View Generation.* With the guidance from factual and counterfactual outcomes, we now can learn an interventional view generator, which aims to generate augmented views by controlling the treatment variable. Given the intermediate node representations $Z$ from our GCL encoder (detailed later) and the treatment indication $T$ or $T^{CF}$, we can generate a homophilic view

---

[1]This substitute is widely used to estimate treatment effects from observational data

with adjacency matrix $\hat{A}$ and a heterophilic view with adjacency matrix $\hat{A}^{CF}$ as follows:

$$\widehat{A} = g_\theta(Z, T), \quad \widehat{A}^{CF} = g_\theta(Z, T^{CF}), \quad (8)$$

where $g_\theta(\cdot, \cdot)$ denotes the view generator, and we empirically adopt a simple multi-layer perceptron:

$$\widehat{A}_{ij} = \text{MLP}_\theta([\mathbf{z}_i \odot \mathbf{z}_j, T_{ij}]), \quad \widehat{A}_{ij}^{CF} = \text{MLP}_\theta([\mathbf{z}_i \odot \mathbf{z}_j, T_{ij}^{CF}]) \quad (9)$$

where $\odot$ represents Hadamard product, and $[\cdot, \cdot]$ denotes the vector concatenation. The value of each entry in $\widehat{A}$ and $\widehat{A}^{CF}$ is in the range of $[0, 1]$. Here is the physical meaning of these two views: the homophilic structure $\widehat{A}_{ij}$ represents the probability of homophily presented between node $v_i$ and $v_j$, and $\widehat{A}_{ij} \to 1$ suggests a strong homophily presence; the heterophilic structure $\widehat{A}_{ij}^{CF}$ represents the probability of heterophily, and $\widehat{A}_{ij}^{CF} \to 1$ indicates a strong heterophily presence between node $v_i$ and $v_j$.

## 4.2 Dual Graph Encoders

As discussed in Section 3.2, the generated homophilic and heterophilic views require tailored graph filters to capture their distinct patterns. Therefore, we apply a low-pass filter to encode the homophilic view and a high-pass filter to capture heterophilic view. Ultimately, we aim to simultaneously capture smoothly varying information in the homophilic view and sharply changing information in the heterophilic graph.

*4.2.1 Low-Pass Encoder.* For the homophilic view that connects similar nodes, we normalize its adjacency matrix $\tilde{A} = \widehat{D}^{-1/2} \widehat{A} \widehat{D}^{-1/2}$, and utilize low-pass graph filters in SGC [32] to smooth node representations as follows:

$$Z_l^{homo} = \tilde{A} \, Z_{l-1}^{homo} \, W_{l-1}^{low}, \quad \text{and } Z_0^{homo} = X, \quad (10)$$

where $l \in \{1, \ldots, L\}$ denotes the layer index, and the final-layer representation matrix for the homophilic view is $Z^{homo} = f_\phi^{low}(X, \widehat{A})$, where $f_\phi^{low}(\cdot, \cdot)$ is the high-pass encoder with parameters $\phi = \{W_l | l \in \{0, \ldots, L-1\}\}$.

*4.2.2 High-Pass Encoder.* To better encode the heterophilic view that connects dissimilar nodes, we employ high-pass filters to diversity node representations, which can be formulated as:

$$Z_l^{heter} = (I - \alpha \tilde{A}^{CF}) \, Z_{l-1}^{heter} \, W_{l-1}^{high}, \quad \text{and } Z_0^{heter} = X, \quad (11)$$

where $\tilde{A}^{CF}$ is normalized $\widehat{A}^{CF}$, and $\alpha$ as a hyperparameter controls the filtering strength. The final heterophilic representations obtained by the high-pass encoder is $Z^{heter} = f_\psi^{high}(X, \widehat{A}^{CF})$.

## 4.3 Model Training

*4.3.1 View Generator Optimization.* We aim to optimize the view generator via two objectives: (1) accurate node pair prediction on observed outcomes and (2) accurate prediction on estimated counterfactual outcomes, which can be defined as the following cross-entropy loss:

$$\mathcal{L}_F = \mathbb{E}_{i,j \sim \mathcal{V}} \left[ A_{ij} \cdot \log \widehat{A}_{ij} + (1 - A_{ij}) \cdot \log(1 - \widehat{A}_{ij}) \right], \quad (12)$$

$$\mathcal{L}_{CF} = \mathbb{E}_{i,j \sim \mathcal{V}} \left[ A_{ij}^{CF} \cdot \log \widehat{A}_{ij}^{CF} + (1 - A_{ij}^{CF}) \cdot \log(1 - \widehat{A}_{ij}^{CF}) \right]. \quad (13)$$

---

**Algorithm 1** GCL-IVG

---

**Input:** adjacency matrix $A$, node feature $X$
1: *# Gather outcome and counterfactual outcome*
2: Pre-compute $T$ based on X, as presented in Section 4.1.2.
3: Pre-compute $T^{CF}$, $A^{CF}$ by Eq. (6) and (7).
4: Initialize view generator $\theta_0$, low- and high-pass encoder $\phi_0$, $\psi_0$.
5: **for** $t = 0, \ldots, N$ **do**
6:     Get node representation $Z_t^{homo}$, $Z_t^{heter}$ by Eq. (10), (11).
7:     *# Optimize view generator*
8:     Generate homo. and heter. view $\widehat{A}_t$ and $\widehat{A}_t^{CF}$ by Eq. (8).
9:     Update view generator $\theta_t$ by Eq. (15).
10:     *# Optimize dual encoders*
11:     Update encoders $\phi_t$ and $\psi_t$ by Eq. (16).
12: **end for**
**Output:** Representations $Z = [Z_N^{homo}, Z_N^{heter}]$, views $\widehat{A}_N$ and $\widehat{A}_N^{CF}$

---

In practice, we regularize the homophilic and heterophilic probabilities by making their summation for each node pair to be one:

$$\mathcal{L}_{\text{reg}} = \mathbb{E}_{i,j\sim\mathcal{V}}||1 - \widehat{A}_{ij} - \widehat{A}_{ij}^{CF}||. \tag{14}$$

Therefore, the combination of $\widehat{A}$ and $\widehat{A}^{CF}$ describe a mixing pattern of homophily and heterophily inferred from the original graph. The final objective for optimizing the view generator $g_\theta(\cdot)$ is given by:

$$\min_\theta \mathcal{L}_F + \mathcal{L}_{CF} + \mathcal{L}_{\text{reg}}. \tag{15}$$

*4.3.2 Dual Encoder Optimization.* We follow a common practice in GCL to employ a InfoNCE contrastive loss in Eq.2 for optimizing the dual encoders $f_\phi^{low}(\cdot, \cdot)$ and $f_\psi^{high}(\cdot, \cdot)$:

$$\min_{\phi,\psi} \mathbb{E}_{i\sim\mathcal{V}, j\in N_i'}\left[\mathcal{L}_{\text{GCL}}(\mathbf{z}_i^{heter}, \mathbf{z}_j^{homo}) + \mathcal{L}_{\text{GCL}}(\mathbf{z}_i^{homo}, \mathbf{z}_j^{heter})\right], \tag{16}$$

where $N_i' = \text{TopK}(v_i)$ is a set of nodes whose representation is similar to node $v_i$. We use them as positive examples to mitigate the negative impact of possible noisy edges in the generated views. The node embeddings $Z$ used for downstream tasks is a concatenation from both branches, i.e., $\mathbf{z}_i = [\mathbf{z}_i^{homo}, \mathbf{z}_i^{heter}]$.

*4.3.3 Training Strategy.* Training our model is essentially is a *bi-level optimization* problem: the optimization of view generator relies on the intermediate node representations given by the dual encoders, i.e., Eq. (8) and Eq. (9); the optimization of dual encoders is based on the generated views, i.e., Eq. (12) and Eq. (13). To solve this bi-level optimization problem, we update the view generator and dual encoders in an *alternating manner*, as Algorithm 1 shows.

## 5 EXPERIMENTS

### 5.1 Experimental Settings

*5.1.1 Datasets.* We evaluate GCL-IVG on nine node classification benchmarks covering heterophilic and homophilic graphs. Tabel 1 summarizes the dataset statistics. Specifically, For graphs with heterophily, we adopt one actor co-occurrence network: Actor [25], three webpage datasets: Cornell, Texas, and Wisconsin [20], and two Wikipedia networks: Chameleon and Squirrel[22]. For graphs

**Table 1: Dataset**

| Dataset | | #Node | #Edges | #Features | #Classes | $H_{\text{edge}}$ | $H_{\text{node}}$ |
|---|---|---|---|---|---|---|---|
| Heterophily | Texas | 183 | 309 | 1703 | 5 | 0.061 | 0.097 |
| | Wisconsin | 251 | 499 | 1703 | 5 | 0.170 | 0.150 |
| | Actor | 7600 | 29926 | 931 | 5 | 0.216 | 0.221 |
| | Squirrel | 5201 | 216933 | 2089 | 5 | 0.223 | 0.216 |
| | Chameleon | 2277 | 36051 | 2325 | 5 | 0.234 | 0.247 |
| | Cornell | 183 | 295 | 1703 | 5 | 0.298 | 0.386 |
| Homophily | Citeseer | 3327 | 9104 | 3703 | 7 | 0.736 | 0.717 |
| | Pubmed | 19717 | 88648 | 500 | 3 | 0.802 | 0.792 |
| | Cora | 2708 | 10556 | 1433 | 6 | 0.810 | 0.825 |

with homophily, we adopt three widely used citation networks: Cora, Citeseer, and Pubmed [11].

*5.1.2 Baselines.* We conduct a comprehensive comparison of GCL-IVG with two distinct groups of methods, namely semi-supervised and unsupervised approaches. For conventional method without using graph structure, we adopt **Logistic Regression (LR)**. For semi-Supervised baselines without considering heterophily, we include: **GCN** [11], **GAT** [29] and **APPNP** [6]. We also consider a set of semi-Supervised baselines tailored for heterophily: **GPR-GNN** [5], **Mix-Hop** [1] and **H2GCN** [43]. As a more close framework, vanilla contrastive learning methods that implicitly works with homophily are considered: **MVGRL** [8], **GRACE** [44] and **BGRL** [28]. Finally, we include the most related contrastive learning frameworks with heterophily: **Selene** [15], **HGRL** [3] and **GREET** [15]. This comprehensive comparison covers a wide spectrum of methods, allowing us to thoroughly assess GCL-IVG's performance against various benchmarks for both homophilic and heterophilic graph structures.

*5.1.3 Evaluation Protocol.* In the context of unsupervised methods, we adopt the transductive setting outlined in [28]. This entails pretraining node representations across the entire graph dataset (without accessing labels), and using obtained representations as features for the downstream node classification task, to evaluate the quality of the acquired node representations. We split all datasets following the public splits [11, 20, 36] and commonly used splits [27, 45]. In unsupervised methods, we apply a logistic regression classifier for the downstream task, thereby obtaining node classification results. Classification accuracy serves as our primary evaluation metric. We conduct GCL-IVG and baseline methods using ten different random splits and report the average classification accuracy and standard deviation for the test nodes.

*5.1.4 Model Hyperparameters and Implementation Details.* In our experimental setup, all experiments were conducted using PyTorch on an Nvidia 3090 GPU to keep the same computational environment. To ensure a fair comparison of overall performance, we fixed the embedding dimension for the node classification task at $d = 512$ for all methods. For baseline methods, we adopted the learning rates reported in their corresponding papers. For our approach, we set the learning rate for optimizing the view generator as 0.001 and the learning rate for optimizing duel encoders as 0.001. When utilizing Node2Vec for counterfactual outcome estimation (i.e., Eq. (6)), we set a fixed random walk length of 24 to acquire structural embeddings. We keep all edges $\mathcal{E}$ and randomly sample $50|\mathcal{E}|$ unconnected node pairs as the observed outcome set to improve the efficiency of calculating counterfactual outcomes.

**Table 2: Node classification accuracy (mean accuracy with standard deviation over different splits) comparison on datasets. $X$, $A$, $Y$ denote node original features, adjacency matrix, and node labels, respectively. "*" indicates that results from the original papers. The best and second best results for each dataset are highlighted in bold and underline. The symbol "OOM" denotes out of memory.**

| Avaliable Data | Method | Texas | Wisconsin | Actor | Squirrel | Chameleon | Cornell | Citeseer | Pubmed | Cora |
|---|---|---|---|---|---|---|---|---|---|---|
| X,Y | LR | 58.36±5.37 | 55.87±7.97 | 35.11±0.77 | 27.62±1.03 | 32.99±1.14 | 56.87±6.01 | 56.75±0.86 | 76.80±0.89 | 54.65±2.04 |
| X,A,Y | GCN | 58.44±4.21 | 51.79±6.03 | 28.17±0.44 | 39.50±1.54 | 54.65±2.17 | 54.76±2.78 | 70.19±1.08 | 85.24±0.14 | 82.26±1.20 |
| X,A,Y | GAT | 56.74±3.06 | 51.00±4.34 | 26.85±0.84 | 41.01±0.91 | 55.18±1.95 | 54.83±3.08 | 70.62±0.82 | 84.71±0.34 | 82.82±0.97 |
| X,A,Y | APPNP | 60.34±5.70 | 63.09±8.37 | 34.00±0.65 | 31.76±0.80 | 45.29±1.93 | 62.04±4.80 | 72.40±0.88 | 87.45±0.19 | 84.83±1.34 |
| X,A,Y | GPRGNN | 60.07±5.22 | 57.91±8.37 | 33.29±0.66 | 35.74±2.35 | 51.93±1.57 | 59.86±6.25 | 72.70±0.77 | 87.62±0.30 | 85.24±1.14 |
| X,A,Y | MixHop | 55.51±3.27 | 51.49±5.39 | 29.04±0.97 | 39.36±1.60 | 53.79±1.19 | 52.79±6.25 | 66.40±1.75 | 84.92±0.460 | 81.01±1.55 |
| X,A,Y | H2GCN* | 84.86±6.77 | 86.67±4.69 | 35.86±1.03 | 37.90±2.02 | 54.02±1.56 | 82.16±4.80 | **77.07±1.64** | **89.59±0.33** | **87.81±1.35** |
| X,A | MVGRL | 61.70±3.94 | 50.64±5.89 | 31.37±0.83 | 33.49±0.84 | 42.34±2.11 | 56.19±2.42 | 71.88±0.71 | OOM | 84.53±1.05 |
| X,A | GRACE | 63.54±2.57 | 53.83±3.56 | 28.14±0.81 | 34.47±1.11 | 45.89±3.10 | 56.39±2.11 | 71.37±0.96 | 77.55±1.01 | 83.69±0.73 |
| X,A | BGRL | 65.78±2.66 | 59.80±4.08 | 29.80±0.31 | 31.50±0.57 | 45.54±1.94 | 56.67±2.13 | 69.81±0.56 | 84.65±0.4 | 83.01±0.710 |
| X,A | Selene | 63.95±1.73 | 55.47±4.79 | 33.15±0.39 | OOM | 38.93±1.44 | 56.05±2.50 | 54.08±1.08 | 81.67±0.25 | 56.19±1.52 |
| X,A | HGRL | 77.69±2.42 | 77.51±4.03 | 36.66±0.35 | 35.42±0.91 | 45.041±1.91 | 77.62±3.25 | 70.89±0.75 | 84.18±0.22 | 82.08±0.84 |
| X,A | GREET* | 87.03±2.36 | 84.90±4.48 | 36.55±1.01 | 42.29±1.43 | 63.64±1.26 | **85.14±4.87** | 73.08±0.84 | 80.29±1.00 | 83.81±0.87 |
| X,A | GCL-IVG | **90.09±4.19** | **87.06±2.54** | **38.67±0.39** | **42.36±0.79** | **67.76±1.9** | 82.70±3.86 | 72.13±0.56 | 86.64±0.26 | 84.81±0.48 |

**Table 3: The edge homophily score of generated views.**

| | Cora | Citeseer | Texas | Wisconsin | Cornell | Chameleon | squirrel |
|---|---|---|---|---|---|---|---|
| $\mathcal{G}$ | 0.810 | 0.736 | 0.061 | 0.170 | 0.298 | 0.234 | 0.223 |
| $\widehat{A}$ | 0.849 | 0.830 | 0.587 | 0.522 | 0.560 | 0.629 | 0.367 |
| $\widehat{A}^{CF}$ | 0.735 | 0.709 | 0.061 | 0.179 | 0.131 | 0.210 | 0.188 |

## 5.2 Experimental Results

*5.2.1 Node classification performance.* As shown in Table 2, we evaluated the effectiveness of our proposed method on node classification tasks, compared to various baselines. We can draw the following conclusions:

- *logistic regression (LR)* using only node raw features showed competitive performance on most heterogeneous graphs, highlighting the importance of node raw features;
- *Supervised methods* performed similarly across all datasets, which could be attributed to the presence of partially unlabeled training samples (i.e., 10 % of nodes), underscoring the significance of labels for supervised methods;
- *Unsupervised methods with homophily assumption* exhibited degraded performance on heterophilic graphs, indicating the necessity for specialized design to accommodate graph heterophily;
- *Unsupervised methods tailored for homophily* in general present appealing performance on both homophilic and heterophilic graphs. However, Selene, performed worse, suggesting weaker generalization of learned representations. HGRL emphasizes the proximity of learned representations to the original node features; therefore, its performance does not show advantages on datasets where node raw features perform moderately.
- *GDM*'s performance on heterophilic graphs significantly outperforms the baseline, while it remains competitive on homophilic graphs. The primary reason for this lies in the fact that traditional contrastive methods continuously smooth the representations along heterophilic edges, rendering the representations indistinguishable. In contrast, our approach can disentangle the homophilic and heterophilic information in the graph, and sharpen

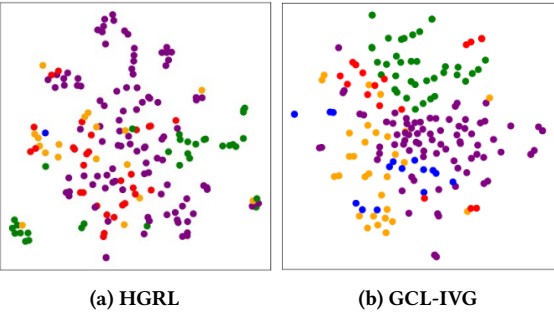

**(a) HGRL**  **(b) GCL-IVG**

**Figure 4: Visualization of node representations obtained by HGRL and our GCL-IVG on Cornell.**

the representation of heterophilic edges through high-pass graph filtering. Consequently, the performance on heterophilic graphs even surpasses that of semi-supervised GNNs. Moreover, due to our counterfactual design, which doesn't simply rely on edge similarity like GREET but instead follows principled guidelines by treatment, we observe that our model learns features with excellent generalization capabilities and higher performance.

*5.2.2 View Generator Performance.* To validate the effectiveness of our view generator, we now analyze the property of generated homophilic views (i.e., $\widehat{A}$) and heterophilic views (i.e., $\widehat{A}^{CF}$). In Table 3, we showcase the graph edge homogeneity scores (i.e., Eq. (4)). Notably, the homophilic scores of the generated homophilic views show sigfinicantly larger homophily score across all datasets. This effect is especially pronounced on heterophilic graphs. Meanwhile, the homophilic scores of the generated heterophilic views are, for the most part, lower than those of the original graphs. This verifies that the view generator indeed serves its role to separate homophilic and heterophilic patterns from the original graph.

It is important to highlight that in the case of the Texas and Wisconsin datasets, the homophilic scores of the generated heterophilic views have actually increased slightly, compared with the original

graph. This anomaly can be attributed to the inherent strength of heterophily within these datasets, which leaves a small room to make it more heterophilic.

*5.2.3 Node Representation Visualization.* Figure 1(a)(b) and 4 visualize the node representations learned by GRACE, HLCL, HGRL and GCL-IVG on the Cornell dataset respectively. Nodes of the same color indicate the same class label. The results indicate that our GCL-IVG is capable of generating discriminative representations where nodes of the same class are cohesively grouped in the same cluster. These findings align with the superior node classification performance of GCL-IVG, demonstrating its effectiveness on heterophilic graphs.

## 5.3 Ablation Study

In this section, we conduct an ablation study to study the gain brought by each design. We consider the following variants:

- w/o $\theta$ Update: the view generator fix its initial random parameters to produce homophilic and heterophilic views. The gradient backpropagation on $\theta$ is block in the training of GCL-IVG.
- w/o $\phi, \psi$ Update: the dual encoders fix its initial parameters, and the backpropagation on $\phi, \psi$ is blocked.
- GRACE: no view generator and dual encoders employed.
- HLCL: only dual encoders are used.
- Homo. Only: only keep the homophilic view and apply both high-pass and low-pass filters for contrastive learning.
- Heter. Only: only keep heterophilic view and apply both high-pass and low-pass filters for contrastive learning.

Results are shown in Table 4, and we have the following observations:

- *It is conspicuously evident that segregating homophilic and heterophilic information from the original graph results in a substantial performance enhancement.* This task goes beyond the scope of conventional unsupervised contrastive methods like HLCL, which rely solely on high-pass and low-pass filter graph encoders. The synergistic use of high and low-pass filters demonstrates superior efficacy in learning from heterophilic graphs, while the opposite holds true for homophilic graphs. This underscores the pivotal significance and effectiveness of disentangling homophilic and heterophilic information within the original graph.
- *The contrastive learning module loss $\mathcal{L}_c$ has a more pronounced impact on the overall loss.* When we individually suspend training for the view generator module and the contrastive learning module, it becomes evident that the loss designs for both modules exert critical influence over the model.
- *An amalgamation of both perspectives in representation emerges as the most auspicious approach, as they encapsulate essential and distinctive insights from varying vantage points.* We shift our focus to assess the contributions of generating two views for each module. Homophilic views are highly effective for homophilic graphs like Cora and CiteSeer, while heterophilic views work better for heterophilic graphs like Texas, Cornell, and Wisconsin.

## 5.4 Parameter Analysis

**High-Pass Filtering Strength $\alpha$ Sensitivity** In this experiment, we vary $\alpha$ from 0.1 to 1 to examine its impact on the model. We

**Table 4: Node Classification Ablation Study Results**

| Ablation | Cora | Citeseer | Texas | Wisconsin | Cornell | Chameleon |
|---|---|---|---|---|---|---|
| GCL-IVG | 84.81 | 72.13 | 90.09 | 87.06 | 82.70 | 66.76 |
| w/o $\theta$ update | 81.51 | 69.11 | 87.01 | 85.86 | 77.11 | 57.45 |
| w/o $\phi, \psi$ update | 63.78 | 52.74 | 80.81 | 76.28 | 69.46 | 54.38 |
| GRACE | 83.90 | 70.91 | 65.45 | 58.82 | 56.76 | 48.22 |
| HLCL | 79.01 | 67.20 | 69.45 | 67.64 | 64.86 | 59.64 |
| Homo. Only | 83.89 | 71.77 | 68.65 | 65.29 | 62.97 | 67.07 |
| Heter. Only | 72.81 | 68.91 | 88.11 | 86.24 | 82.70 | 54.16 |

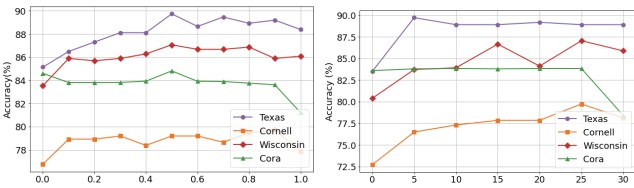

(a) High-Pass Filtering Strength $\alpha$     (b) The number of Top-k

**Figure 5: Parameter sensitivity of $\alpha$ and $k$.**

present the classification accuracy under different $\alpha$ choices in Figure 5(a). From the figures, it can be observed that the best results across these four datasets often occur when $\alpha$ is around 0.5. A common phenomenon is that when $\alpha = 0$, the high-pass filter can be viewed as an MLP, where the central node's neighboring nodes do not participate in message aggregation. Conversely, when $\alpha = 1$, the central node itself does not partake in message aggregation, leading to a decrease in performance.

**The number of TopK Sensitivity** Recall that we retrieve the top $K$ similar nodes as positive examples in contrastive loss. We now investigate the sensitivity of the model to the number of $K$, which ranges from 0 to 30 with a 5-unit interval. Experimental results are shown in Figure 5 (b). It is evident that the optimal choice of $k$ varies for different datasets when $k$ is either too large or too small. We hypothesize that an excessively large $k$ may introduce irrelevant context nodes, thereby increasing misguidance, while an excessively small $k$ may result in insufficient supervisory signals.

## 6 CONCLUSION

In this paper, our primary focus lies in the design of a graph contrastive learning framework, denoted as GCL-IVG. This framework exhibits the capability to accommodate varying degrees of homophily and heterophily within the network structure. To address this challenge, we introduce a counterfactual approach aimed at disentangling heterophilic patterns from homophilic ones. By manipulating the treatment factor, we are able to generate what we term *interventional* homophilic and heterophilic views from the original graph. These two perspectives are then channeled into separate graph filters. We seamlessly integrate the process of interventional view generation with distinct view filters within a conventional contrastive learning pipeline. An extensive array of experiments has been conducted to showcase the outstanding effectiveness and adaptability of our methodology.

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

# A APPENDIX

In this appendix, we present a more in-depth motivational analysis of our method, GCL-IVG. Specifically, we begin by emphasizing the significance of separating the learning of heterogeneous and homogeneous information from the original graph.

## A.1 Performance Variation of GCL Variants on More Datasets

We present the visualization of node representations obtained by a GCL variant in multiple datasets in Section. We demonstrate the widespread effectiveness of separating homophilic and heterophilic information from the original graph. The method of separating these two types of edges leads to improved clustering and node classification performance.

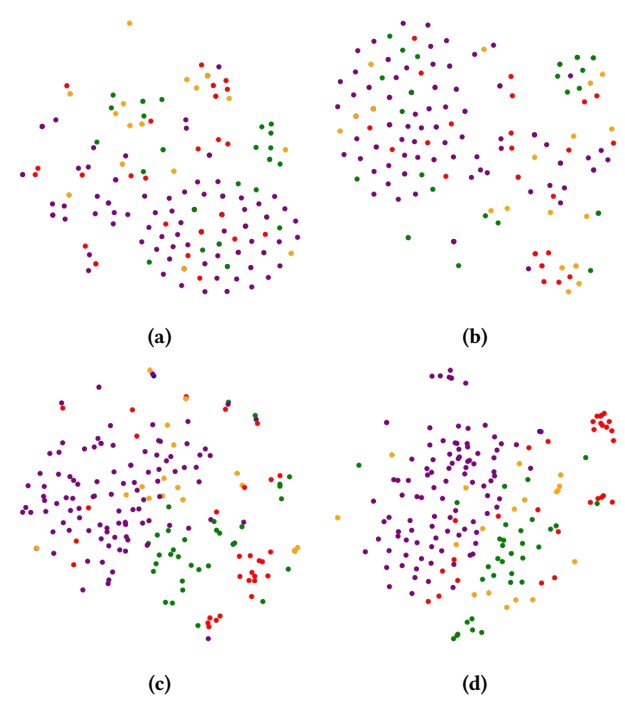

Figure 6: Visualization of node representations obtained by GCL variants on Texas: (a). GRACE; (b). HLCL; (c). $f_{low}(\mathcal{G}^{homo^*}) \oplus f_{low}(\mathcal{G}^{heter^*})$; (d). $f_{low}(\mathcal{G}^{homo^*}) \oplus f_{high}(\mathcal{G}^{heter^*})$.

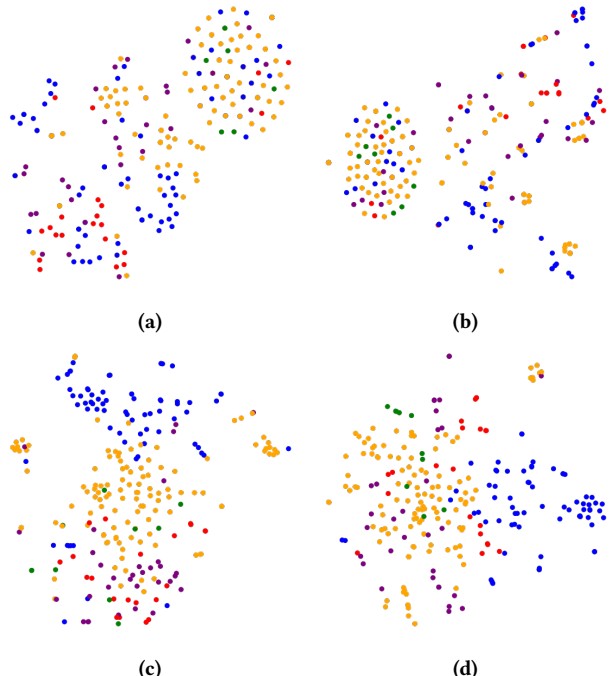

Figure 7: Visualization of node representations obtained by GCL variants on Wisconsin: (a). GRACE; (b). HLCL; (c). $f_{low}(\mathcal{G}^{homo^*}) \oplus f_{low}(\mathcal{G}^{heter^*})$; (d). $f_{low}(\mathcal{G}^{homo^*}) \oplus f_{high}(\mathcal{G}^{heter^*})$.

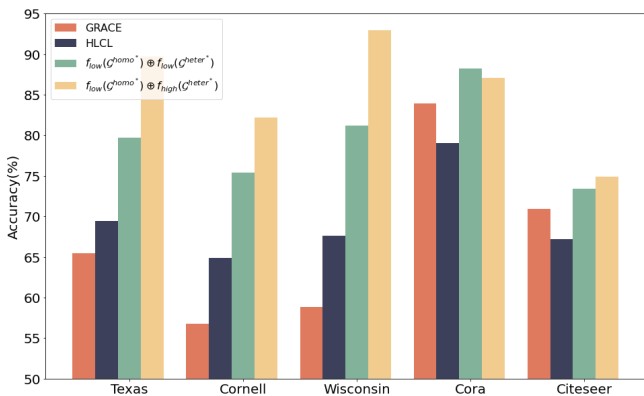

Figure 8: The node classification by different GCL variants

