# OpenReview forum: "Graph Contrastive Learning via Interventional View Generation"
_ACM.org/TheWebConf/2024/Conference — TheWebConf24 Oral_

### Official Review · Reviewer_bVX9 · 2023-11-22

**Novelty:** 4
**Technical Quality:** 4

**Review:**

This paper highlights the existence of heterophilic that connect dissimilar nodes. Thus, the paper generates homophilic and heterophilic views through counterfactual intervention, which targets on disentangling homophilic and heterophilic structure from the original graph. The disentangled graph structures are then channeled into separate graph filters to learn node embeddings in a self-supervised way.

**Questions:**

Strength:
1. The motivation of the paper is interesting and easy to understand. Heterophilic edges are indeed exist in many real-world graph data.
2. The paper presents a preliminary analysis on a heterophilic graph to show case how GCL can be tailored to adapt heterophily. The improved clustering performance also highlight the benefits of segregating homophilic and heterophilic edges, as well as employing specialized low- and high-pass filters.
3. The paper proposes a counterfactual intervention method for graph structure generation. Through controlling the treatment variable T_ij, the model could generate homophilic and heterophilic view of the graph.

Weakness:
1. The counterfactual outcome is defined by prior knowledge, which makes it highly subjective and cannot adapt to all the situations in the real world.
2. The complexity analysis of the model is missing. It is highly recommended to add the complexity analysis of the proposed model.
3. It is better to add more downstream task for comparing, like clustering, graph classification to demonstrate the effectiveness of the model.
4. It is better to add more visualization and case studies to analysis two different representations obtained from two views.

**Reviewer Confidence:**

3: The reviewer is confident but not certain that the evaluation is correct

**Scope:**

4: The work is relevant to the Web and to the track, and is of broad interest to the community

---

### Official Review · Reviewer_DD5t · 2023-11-23

**Novelty:** 4
**Technical Quality:** 3

**Review:**

This paper proposes a graph contrastive learning framework (GCL-IVG) that can handle varying degrees of homophily and heterophily in network structures. The key idea is to disentangle the homophilic and heterophilic patterns from the original graph using a counterfactual approach. Specifically, interventional homophilic and heterophilic views are generated by manipulating a treatment factor. These views are then encoded via separate graph filters before being fed into the contrastive learning pipeline.


Strengths:
1. The proposed method of using counterfactual intervention to generate distinct homophilic and heterophilic views is novel and principled. This allows the contrastive learning framework to capture signals from both perspectives.
2. The integration with existing contrastive learning pipelines is clean without much change to model architecture or learning algorithms. This makes adoption easier.

Weaknesses:
1. This paper proposes a GCL framework for graph heterophily. Its main problem and perspective are graph heterophily, not GCL, which is just a tool used to solve the problem in this paper. So, I don't think the title of the article is appropriate.
2. The method in this paper lacks motivation and inspiration. The technical innovation content of this paper is mainly based on the preliminary analysis in section 3.2. In fact, the preliminary analysis is just a comparative experiment on four variants of "whether to divide homophobic and heterogeneous edges" and "whether to use different filters for different views". Finally, it is found that the fourth variant has the best effect, and then the conclusion is drawn to guide the subsequent framework design. In my opinion, this way of technological innovation lacks motivation and inspiration, and is too crude. In addition, the subsequent use of counterfactual causal inference for interventional view generation also lacks motivation.
3. Lack of simple and powerful baselines. A series of papers [1,2,3] from the perspective of GCL, which do not have any special design for graph heterophily, are simpler than this paper. They not only achieved excellent results on a large number of homophily graphs, but also achieved comparable or better results on heterophily graphs than this paper. This paper lacks discussion and comparison of these methods.

[1] Wang H, Zhang J, Zhu Q, et al. Augmentation-free graph contrastive learning with performance guarantee[J]. arXiv preprint arXiv:2204.04874, 2022.
[2] Single-Pass Contrastive Learning Can Work for Both Homophilic and Heterophilic Graph. Transactions on Machine Learning Research 2023.
[3] Zhang H, Wu Q, Wang Y, et al. Localized Contrastive Learning on Graphs[J]. arXiv preprint arXiv:2212.04604, 2022.
[4] Ning Z, Wang P, Wang P, et al. Graph soft-contrastive learning via neighborhood ranking[J]. arXiv preprint arXiv:2209.13964, 2022.

**Questions:**

Please refer to weaknesses.

**Reviewer Confidence:**

4: The reviewer is certain that the evaluation is correct and very familiar with the relevant literature

**Scope:**

3: The work is somewhat relevant to the Web and to the track, and is of narrow interest to a sub-community

---

### Official Review · Reviewer_GT4h · 2023-11-24

**Novelty:** 5
**Technical Quality:** 6

**Review:**

This paper considers how to adapt graph contrastive learning (GCL) to heterophilous networks. A motivating proof-of-concept shows that separating the graph into homophilous and heterophilous views rather than simply creating random augmented views improves the final performance, and that using a low-pass graph encoder (promoting greater smoothness) for homophilous view but a high-pass encoder for the heterophilous view is even better.  This is what the proposed framework does. However, in practice, in self-supervised learning the node label information can't be used to separate graphs into homophilous and heterophilous components, so causal inference is used to estimate this. A treatment matrix (where positive entries correspond to edge existence between similar node pairs) and a counterfactual treatment matrix (which is the opposite of the treatment matrix) are estimated, and the views are learned as a function of the intermediate node representations from the GNN encoder and the estimated treatment matrix. This becomes a bi-level optimization problem, as the GNN encoder's parameters must also be optimized on that view. (This could conceivably be computationally expensive, and it is worth noting that the graphs used for the experiments are rather small. However, updated experiments and explanation of the computational complexity show that in practice, the method can scale to larger graphs.)

The causal inference perspective on estimating homophilous and heterophilous views is quite interesting. It is worth noting that in some datasets (like Squirrel), the homophilous views are not very homophilous, and in some (as is noted) the heterophilous views are not as heterophilous as the original graphs if they are heterophilous. Some heuristic choices are made in the estimation of the treatment matrices and it would be good to explain these more. Right now, a few parts of this step are also conceptually a little unclear. For instance, how is the counterfactual outcome matrix $A_{CF}$ (eq. 7) used? Learning the views (eq. 8) seems to use the treatment matrices $T$ and $T_{CF}$ to get estimated adjacency matrices / counterfactual outcomes. [update] There is some more helpful discussion of the design choices.

**Questions:**

Q1. Why is a structural encoding needed for the nodes' embeddings, and why node2vec as opposed to, say, Laplacian positional encodings?

Q2. How aggressive is the downsampling for the counterfactual node pairs estimating—how does this affect computational complexity and what are the accuracy tradeoffs?

Q3. How computationally intensive is the bi-level optimization?

**Reviewer Confidence:**

3: The reviewer is confident but not certain that the evaluation is correct

**Scope:**

4: The work is relevant to the Web and to the track, and is of broad interest to the community

---

### Official Review · Reviewer_uUiJ · 2023-11-25

**Novelty:** 4
**Technical Quality:** 4

**Review:**

This paper investigates self-supervised learning on heterophilic graphs. The authors propose that the current general methods for sampling encoders are not suitable for heterophilic graphs. Therefore, a better approach is to divide the graph into two parts, homophily and heterophily, and use low-pass filters and high-pass filters for encoding, respectively. To partition the graph in an unsupervised scenario, this paper proposes a counterfactual learning-based method based on node similarity in the feature space. It defines treatment and counterfactual treatment based on node similarity in the feature space and generates homophilic and heterophily graphs by learning the counterfactual outcome. Then, contrastive learning models are trained on the two generated graphs separately. The experiments show that the proposed method significantly improves performance on heterophily graphs but has less significant effects on homophilic graphs.

Overall, I think this paper has a clear motivation, is sufficiently novel, the methods are reasonable, and it is quite interesting. However, I have some questions about the experimental settings and performance (see below), which prevent me from giving a clear acceptance recommendation. I hope the authors can carefully address my questions, and I would be happy to increase my score if my concerns can be resolved.

**Questions:**

Q1: Is the MLP in equations 8 and 9 the same MLP?

Q2: Where does the high-pass filter defined in Section 4.2.2 come from? I assume there should be a citation here.

Q3: What is the ground truth of $T_{ij}$? It seems to be pre-computed by the similarity between nodes' features, but I guess it would be better if there is a mathematical equation here.

Q4: Why is contrastive loss defined as the formulation in Eq.16? It is weird to me why $z_i^{heter}$ and $z_j^{homo}$ can be viewed as positive pairs. Also, put the detailed loss function here; otherwise, it may cause misunderstanding.

Q5: It is unclear what the detailed split of datasets is. Please specify the split; otherwise, I will not be sure whether the results in Table 2 are reasonable. According to my experience, the results of citation networks are not from the public split. Please do additional experiments of the public split of citation networks.

Q6: The datasets studied in this paper are too small; please consider larger datasets.

Q7: It seems that the method proposed in this paper can be used not only for self-supervised learning. I would like to know if there are similar methods in the supervised learning setting, and if not, I would like to know how this method performs in the supervised learning setting.

Q8: Why is the improvement in the performance of the method proposed in this paper not significant on homophilic graphs? Even in homophilic graphs, there are still many heterophilic edges, so theoretically, this method should also improve performance on heterophilic graphs. If this paper intends to improve performance on heterophilic graphs only, I suggest that the authors modify the title.

Q9: Since $\hat{A}\_{ij}$ and $\hat{A}\_{ij}^{CF}$ are optimized to be close to $A_{ij}$ and $A_{ij}^{CF}$, what's the performance directly using them? How about $T_{ij}$ and $T_{ij}^{CF}$?

**Reviewer Confidence:**

3: The reviewer is confident but not certain that the evaluation is correct

**Scope:**

3: The work is somewhat relevant to the Web and to the track, and is of narrow interest to a sub-community

---

### Decision · Program_Chairs · 2024-01-22

**Decision:**

Accept (Oral)

**Comment:**

This paper attempts to decouple the effects of homophilous and heterophilous parts of a given graph during graph contrastive learning, since the presence of both types of patterns can degrade the representation quality. In order to do so, the paper introduces a counterfactual framework. The proposed method is interesting and novel and is shown empirically to yield good results.

 The reviews also generally recognize the novelty of the proposed method.

 Questions raised by reviews have been addressed during the discussion to a reasonable extent.